# REAL-TIME RISK EVALUATION FOR LLM DECISION-MAKING VIA AN REGRET BOUND

## ABSTRACT

We study real-time risk certification for large language model (LLM) agents with black-box action selection rules, aiming to upper-bound the per-round regret. We fix a reference policy map $f : \mathbb{Y} \to \mathbb{A}$ (e.g., a softmax with temperature $T$, whose TV-Lipschitz constant is $C = \frac{1}{2T}$, though any TV-Lipschitz mapping can be used), which takes a predicted opponent action distribution as input and returns a reference policy. We form the plug-in reference policy $s_{\hat{\mu}_t} = f(\hat{\mu}_t)$ from the model's predicted opponent distribution $\hat{\mu}_t$. Our certificate is $r_t \leq L\big(E_{\text{pred}} + E_{\text{pol}} + E_{\text{mis}}\big)$, where $E_{\text{pred}} := \frac{C}{2}\|\mu_t - \hat{\mu}_t\|_1$ (prediction error), $E_{\text{pol}} := \frac{1}{2}\|\pi_t^* - s_{\mu_t}\|_1$ (policy error), $E_{\text{mis}} := \frac{1}{2}\|\pi_t - s_{\hat{\mu}_t}\|_1$ (policy mismatch), $L$ is the Lipschitz constant of the instantaneous regret with respect to total variation induced by $Q$ (hence domain-dependent), $C$ is the TV-Lipschitz constant of $f$, $\pi_t^*$ denotes the one-hot best response to $\mu_t$ under $Q_t$ (ties broken arbitrarily), and $\pi_t$ is the agent's policy. We assume access at time $t$ to the realized opponent distribution $\mu_t$ and the per-round payoffs $Q_t$ (and hence $\pi^*$), so the certificate is fully computable in real time. In this bound, prediction error measures the accuracy of the model's opponent modeling (belief calibration). In contrast, policy error, together with the policy mismatch $\frac{1}{2}\|\pi_t - s_{\hat{\mu}_t}\|_1$, quantifies the precision of the decision side given $\hat{\mu}_t$. Therefore, this bound enables us to localize the risk of the decision to either prediction or action selection. We applied the certificate to separate, in real time and for black-box policy agents, whether decision risk stems from prediction or from action selection. In the Ultimatum and $2 \times 2$ general-sum games, the dominant component is opponent- and game-dependent. This separation does not yield a characterization common to all games and opponents, but under the same game and opponent strategy, it reveals consistent differences between models.

## 1 INTRODUCTION

LLMs are increasingly deployed as agents that interleave reasoning with tool use and environment interaction. Frameworks such as ReAct (Yao et al. (2023)), Reflexion (Shinn et al. (2023)), and AutoGen (Wu et al. (2024)) have pushed capabilities, yet practical evaluation still leans on aggregate scores that obscure why an agent fails at a particular decision round. This lack of transparency in failure attribution limits the reliability and interpretability of LLM agents in strategic settings, especially when deployed in high-stakes or interactive environments.

We introduce a per round regret certificate that decomposed decision risk into (i)prediction error over opponent behavior, (ii)policy mismatch between the deployed and a reference policy, and (iii)an policy error. The decomposition yields real-time diagnostics that identify whether failures arise from opponent modeling or action selection given the belief. Unlike prior metrics that aggregate performance, our certificate enables per-round attribution of decision risk, revealing whether failures stem from flawed belief calibration or suboptimal action selection.

Across $2 \times 2$ general-sum games and Ultimatum games with LLMs, the dominant term shifts with model, game, and opponent. These results argue for per-round online certification as a practical companion to deployment. This decomposition not only enables real-time attribution of decision risk, but also reveals model-specific weaknesses—whether in belief calibration or action selection—across strategic environments. As such, it offers a practical diagnostic tool for deploying LLM agents in interactive tasks.

## 2 RELATED WORK

### 2.1 LLM AND REGRET IN REPEATED DECISION-MAKING

Early studies evaluate whether LLM agents exhibit no-regret behavior in online learning and games. Park et al. (2025) introduce regret-loss and report settings in which LLMs fail to achieve no-regret, even while equilibria sometimes emerge in repeated play; their metrics are average and cumulative and do not decompose causes regret during play.

Complementary lines formalize regret dynamics and links to equilibria (e.g., regret matching leading to correlated equilibrium), establishing the classical recipe for per-round bounds used in online learning Hart & Mas-Colell (2000) Hart & Mas-Colell (2003).

### 2.2 LLM AS A STRATEGIC DECISION MAKER IN GAMES

Behavioral game-theorem evaluations show that LLM can perform well in self-interest games but struggle with coordination and convention formation. Akata et al. (2025) run large batteries of finitely repeated $2 \times 2$ games (LLM vs LLM and LLM vs human), finding strong performance in Prisoner's Dilemma, brittle coordination in Battle of the Sexes, and sensitivity to prompting and role framing. However, they focus on payoff rates rather than per-round regret causes.

Lorè & Heydari (2024) demonstrate that contextual framing shifts strategies across social dilemmas (PD, Stag Hunt, Snowdrift), underscoring the role of prompt design. However, an evaluation of the decision-making risk via regret is not provided.

Negotiation or ultimatum style settings probe fairness and acceptance thresholds. Recent studies (Polachek et al. (2025), MURASHIGE & ITO (2025), and Guo (2023)) vary stakes and tool stacks (e.g., the use of frameworks like AutoGen Wu et al. (2024)), but focus on behavioral outcomes such as acceptance rates and split distribution, rather than analyzing the underlying risk of decision-making through formal metrics such as a decomposed regret upper bound per-round.

Other studies have pushed for a more nuanced evaluation beyond simple payoffs or win rates. For example, the work on social deduction games provides a fine-grained thematic analysis of LLM behavior (e.g., deception or logical consistency Kim et al. (2024)). Although these qualitative insights are important, they do not offer a quantitative framework for diagnosing the risk of decisions, which requires a rigorous metric like a decomposed per-round regret.

### 2.3 ONLINE LEARNING FOUNDATIONS USED BY LLM-AGENT WORK

The standard toolkit for online learning offers per-round upper bounds on instantaneous regret. These bounds are typically expressed in terms of the payoff range, the distance between the agent's policy and a reference (e.g., softmax), and the discrepancy between the predicted and observed actions of the opponent. This formulation connects regret minimization to equilibrium concepts and enables the diagnosis of performance limitations, attributing them to either a flawed policy or imprecise opponent modeling. Our work adapts this classical scaffold, instantiating it for LLM agents engaged in sequential interactions Hart & Mas-Colell (2000), Shalev-Shwartz et al. (2012).

### 2.4 POSITIONING OF THIS WORK

Across the above, two gaps remain: (i) real-time (per-round) regret control usable during play, and (ii) per-round cause attribution that separates performance limits due to prediction versus action selection (relative to a reference). We address both by deriving an instantaneous upper bound that splits into (a) a prediction term driven by the gap between the realized opponent distribution $\mu_t$ and the model's prediction $\hat{\mu}_t$, (b) an policy mismatch measuring the deviation of the executed policy $\pi_t$ from the reference $s_{\hat{\mu}_t}$, and (c) an optional reference-design term comparing $s_{\mu_t}$ to the best response $\pi_t^*$. We work under a standard-access setting where, at time $t$, $Q_t$, $\mu_t$, and $\pi_t$ are available, so all quantities are able to evaluate online. Crucially, computing these components reveals whether per-round decision risk stems from prediction or from action selection for each model.

TAKEAWAY

Previous work documents nontrivial regret and framing sensitivity in interactive LLM tasks and provides per-round bounds in online learning. We operationalize these tools for LLMs in games by deriving a computable per-round regret upper bound with a three-way decomposition—prediction error, policy mismatch, and (optional) policy error—enabling real-time diagnosis of whether weaknesses stem from opponent modeling or action selection.

## 3 SETUP & NOTATION

We define the total variation $d_{TV}(p, q) = \frac{1}{2}||p - q||_1$. We use $|| \cdot ||_1$ as the vector norm. For a one-hot vector $\delta_y$, we have $d_{TV}(\delta_y, p) = 1 - p(y)$ in the discrete case.

### 3.1 ACTIONS, POLICIES, AND OPPONENT DISTRIBUTION

In this study, we assume finite and discrete action spaces for both the agent and the opponent. The agent chooses an action from a finite set $\mathbb{A} = \{a_1, a_2, ..., a_K\}$, where $K = |\mathbb{A}|$. It implemented decision rule in round $t$ is a probability distribution $\pi_t$ over $\mathbb{A}$; sampling from $\pi_t$ produces the action $a_t$. $\pi_t^*$ is the agent's policy which maximizes the action value we define afterward. In deployment, when $\mu_t$ or $Q_t$ is unavailable. we replace $\pi_t^*$ with $\hat{\pi_t^*} = BR(\hat{\mu}_t, \hat{Q}_t)$.

The opponent then takes an action $y_t$ in an opponent action set $\mathbb{Y}$. We model the opponent by a conditional action distribution $\mu_t(\cdot|a)$ over $\mathbb{Y}$: after the agent plays $a_t$, the opponent's realized action satisfies $y_t \sim \mu_t(\cdot|a)$. The agent also provides a predicted opponent distribution $\hat{\mu}_t$ for all $a \in \mathbb{A}$.

The agent's internal decision rule mapping is unknown and is denoted $f_*$ (opponent policy $\mu_t \to$ agent policy $\pi_t$). For analysis, we fix a reference rule $f$ and form the reference policy $s_{\mu_t} := f(\mu_t)$. This reference rule is a comparator built from observables; it does not need to match $f_*$. We assume $f$ is TV-Lipschitz with constant $C$.

Define $Softmax(x) = \frac{e^{x_i/T}}{\sum_{i \in [d]} e^{x_i/T}}$ for $x$ in $\mathbb{R}^d$, where $T$ is temperature.

### 3.2 PAYOFF AND ACTION VALUES

$R(a, y)$ means one-step payoff. From this, we define the action value $Q_t = \mathbb{E}_{y \sim \mu_t}[R(a, y)]$. Then, we define per-round regret $r_t = EV(\pi_t^*, \mu_t) - EV(\pi_t, \mu_t)$, where $EV(\pi_t^*, \mu_t) = \max_a Q_t(a)$, and $EV(\pi_t, \mu_t) = \sum_a \pi_t(a)Q_t(a)$.

Let $L := \max_y(\max_a R(a, y) - \min_a R(a, y))$. Then, for every action $a$ and the opponent's policy $\mu, \hat{\mu}$,

$$|Q_t(a; \mu) - Q_t(a; \hat{\mu})| \leq L \, d_{TV}(\mu, \hat{\mu}). \tag{1}$$

Moreover, for any $\mu$, $\max_a Q_t(a; \mu) - \min_a Q_t(a; \mu) \leq L$. In particular, $Q_t(a^*) - Q_t(a) \leq L$, because $\max_a Q_t(a; \mu) \geq Q_t(a^*, \mu)$ holds.

## 4 PER-ROUND REGRET CERTIFICATE

Our purpose is to calculate regret upper bound in real time with in a discrete action space.

As above, we define per-round regret $r_t = EV(\pi_t^*, \mu_t) - EV(\pi_t, \mu_t)$. At first we divide this regret into 3 parts.

$$r_t = (EV(\pi_t^*, \mu_t) - EV(s_{\mu_t}, \mu_t)) + (EV(s_{\mu_t}, \mu_t) - EV(s_{\hat{\mu}_t}, \mu_t)) + (EV(s_{\hat{\mu}_t}, \mu_t) - EV(\pi_t, \mu_t)) \tag{2}$$

In the following, we derive the upper bound of each parts in equation 2.

### 4.1 POLICY ERROR

We derive the upper bound of $EV(\pi_t^*, \mu_t) - EV(s_{\mu_t}, \mu_t)$.

At first, $EV(\pi_t^*, \mu_t) - EV(s_{\mu_t}, \mu_t) \leq L d_{TV}(\pi_t^*, s_{\mu_t})$ holds. We define $d_{TV}(p, q) = \frac{1}{2}||p - q||_1$,

so $d_{TV}(\pi_t^*, s_{\mu_t}) = \frac{1}{2}||\pi_t^* - s_{\mu_t}||_1$ is.

Therefore,

$$EV(\pi_t^*, \mu_t) - EV(s_{\mu_t}, \mu_t) \le L\frac{1}{2}||\pi_t^* - s_{\mu_t}||_1 \tag{3}$$

holds, and we refer to this item as policy error.

Policy error means how precisely can the agent selects action in terms of maximizing payoff it gets if they can accurately predict the opponent's policy.

### 4.2 PREDICTION ERROR

$EV(f(\mu_t), \mu_t) - EV(f(\hat{\mu}_t), \mu_t) \le Ld_{TV}(f(\mu_t), f(\hat{\mu}_t))$ holds, because we assume that the action value $Q$ is TV-Lipschitz. In addition, we assume $f$ is TV-Lipschitz with constant $C$, so $d_{TV}(f(\mu_t), f(\hat{\mu}_t)) \le Cd_{TV}(\mu_t, \hat{\mu}_t)$ holds.

Thus, we have

$$EV(f(\mu_t), \mu_t) - EV(f(\hat{\mu}_t), \mu_t) \le LCd_{TV}(\mu_t, \hat{\mu}_t) = LC||\mu_t - \hat{\mu}_t||_1 \tag{4}$$

We refer to $LCd_{TV}(\mu_t, \hat{\mu}_t)$ as prediction error.

Prediction error means how accurately can the agent predict the opponent's action.

### 4.3 POLICY MISMATCH

From the Lipshitz assumption on $Q_t$, it follows that $EV(f(\hat{\mu}_t), \mu_t) - EV(\pi_t, \mu_t) \le Ld_{TV}(f(\hat{\mu}_t), \pi_t)$. Thus, the equation below holds.

$$EV(f(\hat{\mu}_t), \mu_t) - EV(\pi_t, \mu_t) \le Ld_{TV}(f(\hat{\mu}_t), \pi_t) = L\frac{1}{2}||f(\hat{\mu}_t) - \pi_t||_1 \tag{5}$$

We refer to $L\frac{1}{2}||f(\hat{\mu}_t) - \pi_t||_1$ as policy mismatch.

Policy mismatch means how much deferent $f$ and the agent's policy $\pi_t$. When the agent's policy is black-box, we have to tentatively assume $f$ to analysis. However, these are not the same for the most part, so we need to consider this mismatch.

### 4.4 PER-ROUND REGRET UPPER BOUND

From the above, we get regret upper bound

$$r_t \le L(\frac{1}{2}||\pi_t^* - s_{\mu_t}||_1 + C\frac{1}{2}||\mu_t - \hat{\mu}_t||_1 + \frac{1}{2}||f(\hat{\mu}_t) - \pi_t||_1) \tag{6}$$

We are able to compute this bound 6 in real-time (only per-round values).

By examining the dominant term in the three-term decomposition, we can determine whether the model's errors stem from mispredicting the opponent's behavior or from action selection.

More specifically,

(i) prediction error means how exactly the model predicts the opponent's action.

(ii) policy mismatch and policy error mean how accurate the model select their own action.

## 5 EXPERIMENTS

In order to prove our theorem, we conduct experiments in which LLMs (GPT4o, GPT5, gemini2.5 flash lite) play ultimatum games, and $2 \times 2$ general-sum games.

For all experimental settings the regret upper bound coverage rate is 100%.

### 5.1 $2 \times 2$ GENERAL-SUM GAME

In this study, as a $2 \times 2$ general-sum game we use prisoner's dilemma. More details, refer to B, and D. We adopted random and tit-for-tat (TFT) as the opponent's strategies. In this setup, it is providing actions from previous rounds.

PRISONER'S DILEMMA

Prisoner's dilemma is a two player game where defection strictly dominates coopration, so the Nash equilibrium is $(D, D)$ even though mutual cooperation $(C, C)$ would make both better off.
At first, we show a regret and upper bound. in figure 1 to 6. As LLMs, we use GPT4o, gemini2.5 flash lite (temperature: 0.2, 0.5, 0.8), and GPT5.

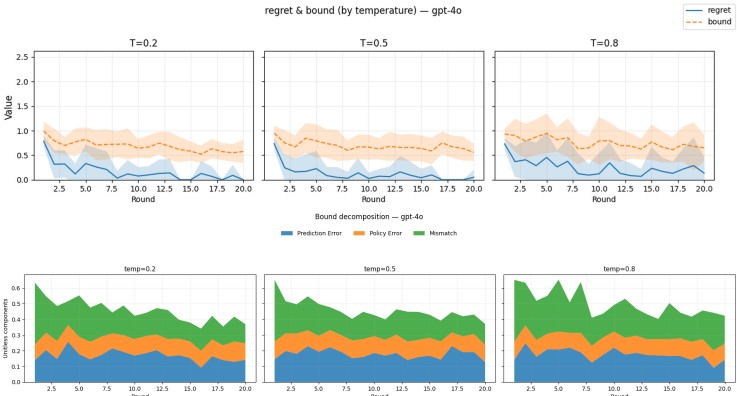

Figure 1: GPT4o's regret and upper bound in prisoner's dilemma; opponent:random (episode:20, round:20)

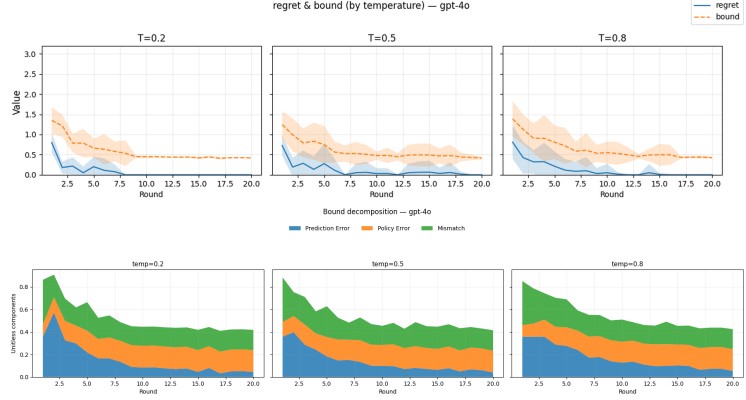

Figure 2: GPT4o's regret and upper bound in prisoner's dilemma; opponent:tft (episode:20, round:20)

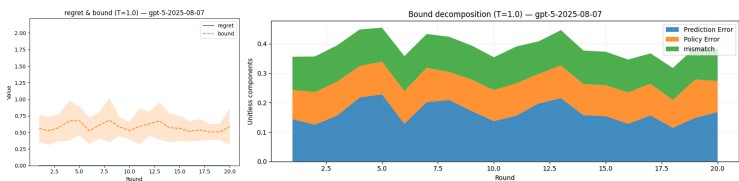

Figure 3: GPT5's regret and upper bound in prisoner's dilemma; opponent:random (episode:20, round:20)

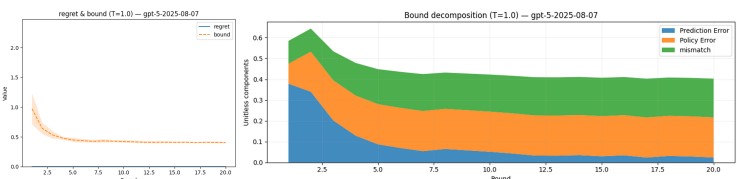

Figure 4: GPT5's regret and upper bound in prisoner's dilemma; opponent:tft (episode:20, round:20)

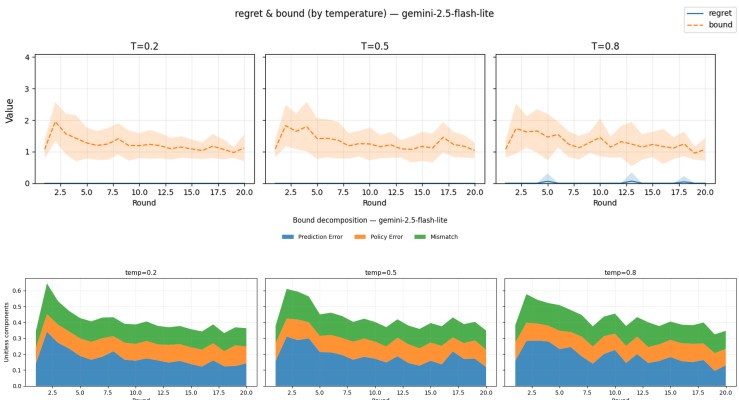

Figure 5: gemini2.5 flash lite's regret and upper bound in prisoner's dilemma; opponent:random (episode:20, round:20)

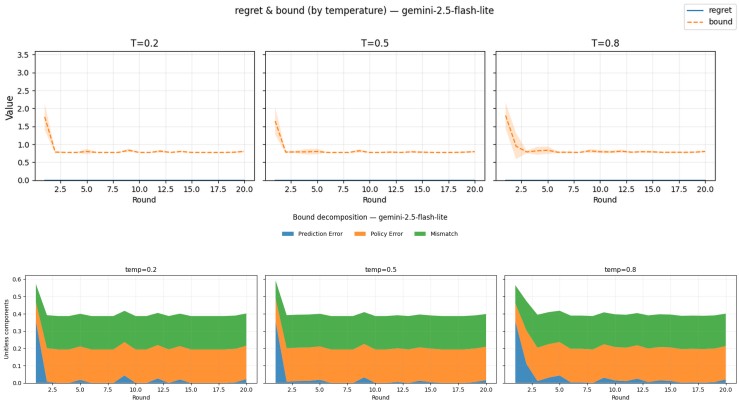

Figure 6: gemini2.5 flash lite's regret and upper bound in prisoner's dilemma; opponent:tft (episode:20, round:20)

In the prisoner's dilemma, LLMs tends to choose defect more frequently than cooperate (90% to 100%). This tendency suggests that the models often adopt a short-term payoff–maximizing strategy rather than pursuing long-term mutual cooperation. In figure 3 to 6, regret is always 0.

Across all models, prediction error decreased due to differences in strategy. Tit-for-Tat(TFT) is a strategy that selects the opponent's action from their previous round. LLMs tend to choose defect, so TFT opponent choose defect more frequently. Thus, models can predict the opponent's actions more accurately, and prediction error becomes smaller as the rounds progress. Similar patterns were observed in other 2×2 general-sum games such as win-win, chicken, and biased.

## 5.2 ULTIMATUM GAME

Ultimatum game is a two-player bargaining game where the proposer offers a split of a fixed pie and responder either accepts or rejects.

In this setup, we model the Ultimatum Game with an LLM as the proposer and a simulated responder. Each round the proposer offers $o \in \{0, 10, ..., 100\}$ from a 100 unit pie. If accepted, payoffs are $(100 - o, o)$; otherwise both receive 0.

We conducted ultimatum game experiments using three LLMs (GPT4o, GPT5, gemini2.5-flash-lite). For GPT4o and gemini2.5-flash-lite, we considered temperature settings of 0.2, 0.5, and 0.8. For the opponent's behavior, we fix the responder's acceptance function $\phi(o)$ over offers $o \in \{10, 20, 30, 40, 50, 60, 70, 80, 90\}$ as $\phi(o) = (0.00, 0.05, 0.10, 0.20, 0.30, 0.20, 0.10, 0.05, 0.00)$.

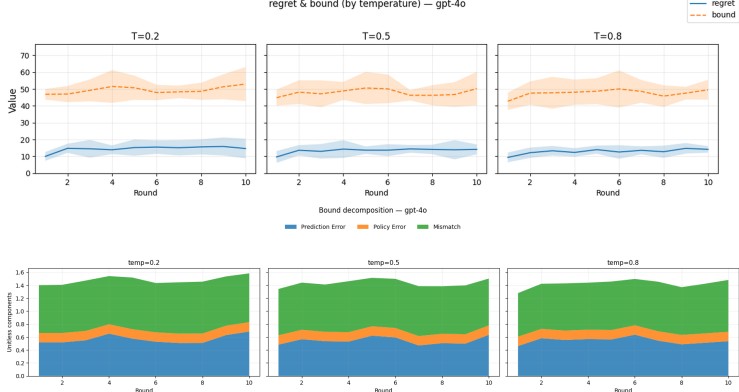

Figure 7: GPT4o's regret and upper bound with 3 decomposed terms in ultimatum game (episode:20, round:20)

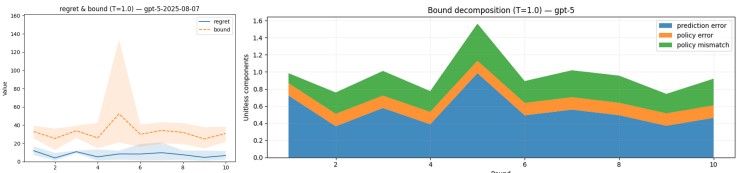

Figure 8: GPT5's regret and upper bound with 3 decomposed terms in ultimatum game (episode:20, round:20)

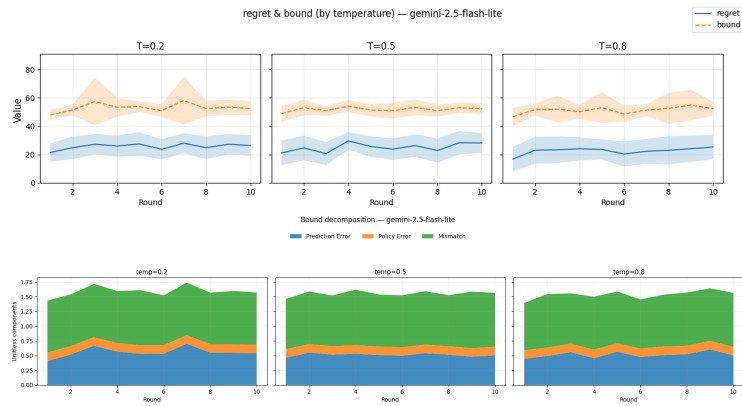

Figure 9: gemini2.5 flash lite's regret and upper bound with 3 decomposed terms in ultimatum game (episode:20, round:20)

Figure 7 to 9 show the regret and upper bound. In GPT4o and gemini2.5 flash lite, the trend of the three upper-bound decomposition term is the same at all temperatures. GPT5 tends to have smaller policy error and policy mismatch compared to GPT4o and gemini2.5 flash lite. This indicate that GPT5 has higher action rationality with residual gaps largely attributable to prediction error. All model have large prediction error, and this means that predict the opponent's policy is very difficult.

## 6 CONCLUSION

We propose a per-round regret upper bound for LLM agents and it can be broken down into three components: prediction error, policy error, and policy mismatch. This decomposition provides per-round attribution of decision risk.

The dominant component shifts across models and tasks, indicating that aggregate scores alone are insufficient. The decomposition specifies whether to improve opponent-belief calibration or to adjust action selection relative to the reference policy. Because the regret certificate is computed in real time, it enables online and targeted interventions without interrupting the interaction. However, the current approach is limited to finite action spaces and single-step payoffs. Overall, our results support the use of per-round certification as a practical tool for deploying LLM agents more safely and effectively.

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

# A  WHY $C = 1/(2T)$ FOR SOFTMAX AND TIGHT

Let $T > 0$ and $f : \mathbb{R}^k \to \Delta_{k-1}$ be $f(z) = \text{softmax}(z/T)$. For total variation $d_{\text{TV}}(x, y) = \frac{1}{2}\|x - y\|_1$, we show the following.

$$d_{\text{TV}}\big(f(x), f(y)\big) \ \leq \ \frac{1}{2T}\, d_{\text{TV}}(x, y) \qquad (\forall x, y \in \mathbb{R}^k),$$

and that the constant $1/(2T)$ is tight.

## A.1  JACOBIAN AND OPERATOR NORM

The Jacobian of softmax is

$$J(z) = \frac{1}{T}(Diag(p) - pp^\top) \tag{7}$$

Martins & Astudillo (2016), Gao & Pavel (2017), where $p$ is $f(z)$.
For each column $j$, $\sum_i |J_{ij}| = \frac{2}{T}\, p_j(1 - p_j) \leq \frac{1}{2T}$, hence $\|J(a)\|_{1\to 1} \leq \frac{1}{2T}$, since $x(1-x) \leq 1/4$.
Hence, the induced norm satisfies

$$\|J(z)\|_{1\to 1} \ = \ \max_j \sum_i |J_{ij}| \ \leq \ \frac{1}{2T}. \tag{8}$$

## A.2  MEAN VALUE FORM

Let $x, y \in \mathbb{R}^k$, $x(t) = x + t(y - x)$ and $g(t) = f(x(t))$. By the chain rule $g'(t) = J(x(t))(y - x)$, and thus

$$f(y) - f(x) = \int_0^1 J\big(x + t(y - x)\big)\,(y - x)\,dt. \tag{9}$$

Taking $\ell_1$ norms and using equation 8 gives

$$\|f(y) - f(x)\|_1 \ \leq \ \int_0^1 \|J(\cdot)\|_{1\to 1}\,dt\,\|y - x\|_1 \ \leq \ \frac{1}{2T}\,\|y - x\|_1. \tag{10}$$

Since $d_{\text{TV}} = \frac{1}{2}\|\cdot\|_1$,

$$d_{\text{TV}}\big(f(x), f(y)\big) \ \leq \ \frac{1}{2T}\, d_{\text{TV}}(x, y). \tag{11}$$

Therefore, $C \leq 1/(2T)$.

## A.3  TIGHTNESS

For $k \geq 2$, fix two coordinates and keep the others constant (equivalently, take $x_{3:k} \to -\infty$ so that $p \to (1/2, 1/2, 0, \ldots, 0)$). Take $x_1 = x_2$ and perturb along $\delta = (\varepsilon, -\varepsilon, 0, \ldots, 0)$. Then $p_1(\varepsilon) = \sigma\big((x_1 - x_2 + 2\varepsilon)/T\big)$ with $\sigma'(0) = 1/4$, so $\frac{d}{d\varepsilon} p_1(\varepsilon)\big|_{\varepsilon=0} = \frac{2}{T} \cdot \frac{1}{4} = \frac{1}{2T}$. Because $d_{\text{TV}}(f(x + \delta), f(x)) = \frac{1}{2}\big(|\Delta p_1| + |\Delta p_2|\big) = |\Delta p_1|$ (the other coordinates are unchanged),

$$\frac{d}{d\varepsilon}\, d_{\text{TV}}\big(f(x + \delta), f(x)\big)\bigg|_{\varepsilon=0} = \frac{1}{2T}, \qquad \frac{d}{d\varepsilon}\, d_{\text{TV}}(x + \delta, x)\bigg|_{\varepsilon=0} = 1,$$

so, the ratio reaches $1/(2T)$ in the small-step limit. Hence, the constant is tight.

# B  DEFINITIONS AND EXAMPLES OF $2 \times 2$ GENERAL-SUM GAMES

In $2 \times 2$ general-sum games, the player $i \in \{1, 2\}$ has action space $A_i = \{C, D\}$ (labels are conventional). Payoffs in the games are $u_i(a_1, a_2)$, where $a_i \in \mathbb{A}_i$ is. The four outcomes map as follows: $(C, C) \to R, (C, D) \to S, (D, C) \to T, (D, D) \to P$.

WIN-WIN

This payoff matrix is characteristic of a win-win game, where the outcome of mutual cooperation $(C, C)$ maximizes both the individual payoffs for each player and the total collective payoff. An example of payoff matrix in win-win game is

$$A = B = \begin{pmatrix} 4 & 3 \\ 2 & 0 \end{pmatrix}$$

Here, $C$ is strictly dominant for both players; the unique Nash equilibrium $(C, C)$ is also Pareto-efficient Osborne & Rubinstein (1994). The alignment between individual rational choices and the collectively optimal outcome is why this structure is colloquially known as a win-win game.

PRISONER'S DILEMMA(PD)

This game features a clash between individual rationality and collective efficiency. Mutual cooperation $(C, C)$ yields a higher joint payoff than defection $(D, D)$, yet defection is strictly dominant for each player. Hence the unique Nash equilibrium is $(D, D)$, which is Pareto-dominanted by $(C, C)$ Osborne & Rubinstein (1994).
An example payoff matrix is

$$A = B = \begin{pmatrix} 5 & 3 \\ 1 & 0 \end{pmatrix}$$

Conditions is $T > R > P > S$ for both players, and $2R > T + S$.

CHICKEN

Chicken is an anti-coordination structure. Each player prefers to choose the opposite of the other. There are two asymmetric equilibria $(C, D)$ and $(D, C)$ and typically one interior mixed equilibrium which outcome emerges depends on conventions or pre-play communication Osborne & Rubinstein (1994), Robinson & Goforth (2005).
An example payoff matrix is

$$A = B = \begin{pmatrix} 3 & 2 \\ 4 & 0 \end{pmatrix}$$

Efficient play often looks like turn-taking. Self correcting memory-1 strategies (e.g., WLSL) help stabilize alternation under noise, whereas unforgiving rules can produce inefficient flare-ups Nowak (2006), Nowak & Sigmund (1993).

UNFAIR

Both players prefer to match actions, but payoffs are tilted toward one player across outcomes. Two equilibria, $(C, C)$ and $(D, D)$, exists. However, the advantaged player receives more in either case, so distributional asymmetry drives the analysis (Robinson & Goforth (2005), Osborne & Rubinstein (1994)). An example payoff matrix is

$$A = \begin{pmatrix} 4 & 0 \\ 6 & 1 \end{pmatrix}, B = \begin{pmatrix} 3 & 0 \\ 5 & 1 \end{pmatrix}$$

In repeated interaction, outcomes tend to favor the advantaged side unless explicit fairness mechanisms, correlation devices, or bargaining protocols counterbalance the asymmetry.

BIASED

Both players want to coordinate, but they disagree about which coordinated outcome is better. There are two equilibria $(C, C)$ favored by the Row player and $(D, D)$ favored by the Column player. Selection among equilibria often reflects focal points, communication, or bargaining power (Osborne & Rubinstein (1994),Robinson & Goforth (2005)). An example payoff matrix is

$$A = B = \begin{pmatrix} 4 & 1 \\ 3 & 2 \end{pmatrix}$$

In repeated play, cheap talk or conventions can steer the pair to one of the coordinated outcomes.

# C SUMMARY OF STRATEGIES IN $2 \times 2$ GENERAL-SUM GAMES

## C.1 PRELIMINARIES

Let $H$ denote the set of finite public histories, and write $h_t \in H$ for the history up to time $t$, e.g., $h_t = ((a_1^1, a_2^1), \ldots, (a_1^{t-1}, a_2^{t-1}))$. A strategy for the player $i$ is a mapping from the history up to time $t$, $h_t$, to a distribution over actions $\sigma_i : h_t \rightarrow \Delta(\mathbb{A}_i)$. We distinguish strategies whether they are deterministic or stochastic, and how much of the past they condition on "memory depth". For a broader context of cooperation mechanisms beyond direct reciprocity, see Nowak (2006).

## C.2 DETERMINISTIC STRATEGIES

A deterministic strategy chooses actions with probability 0 or 1 for every round. Basic memory-0 baselines are **All-C** or **All-D**. Standard history dependent strategies include **Tit-for-Tat(TFT)**(play the opponent's previous action; first move typically $C$). TFT became prominent after computer tournaments in Axelrod (1980a), Axelrod (1980b) on the iterated Prisoner's Dilemma demonstrated its strong performance. Subsequently, these results were framed as evidence for direct reciprocity in Axelrod & Hamilton (1981). **Win-Stay**, and **Lose-Shift**(WSLS/Pavlov), repeat after a successful outcome and switch otherwise, was shown to outperform TFT under noise in the Prisoner's Dilemma Nowak & Sigmund (1993). **Grim Trigger**(defect forever after any observed defection) is formally memory-$\infty$, but is implementable with a single violation flag in a description of finite states (Finite-state / automatic representations of repeated games are standard; see, e.g., textbook treatments building on Kuhn (1953)'s extensive form foundations).
Deterministic strategies are transparent but can be brittle under execution noise: TFT is prone to long chain of retaliation, while WSLS tends to self-correct Nowak & Sigmund (1993).

## C.3 STOCHASTIC STRATEGIES

A stochastic strategy outputs an action distribution, useful for robustness, forgiveness, or deliberate randomization. Memory-1 strategies are especially convenient. Let the previous result be $s_{t-1} \in \{CC, CD, DC, DD\}$, and a policy is the vector $p = (p_{CC}, p_{CD}, p_{DC}, p_{DD})$ with $p_{xy} = Pr(C|s_{t-1} = xy)$. Deterministic strategies are exactly the vertices $\{0, 1\}^4$ of this simplex. Representative stochastic strategies include **Generous TFT**, forgive after $CD$ or $DD$, and **stochastic WSLS** (success dependent transition probabilities). Generous variants are supported by evolutionary analyses showing how generosity can replace extortionary behavior Stewart & Plotkin (2013).

## C.4 ZERO-DETERMINANT STRATEGIES

Zero-determinant (ZD) strategies form a parametric subset of memory-1 that can linearly constrain long-term payoff relations in the repeated game Press & Dyson (2012). Subsequent work shows that extortionary ZD strategies are not evolutionarily stable in large, well-mixed populations. They tend to give way to more reciprocal or generous types Hilbe et al. (2013).

## C.5 WHEN TO USE WHAT

We summarize the traits of these strategies in Table 1.
With nontrivial noise or heterogeneous opponents, memory-1 strategies with forgiveness or self-correction tend to restore cooperation after errors and are practically stable, whereas purely deterministic TFT is more fragile to error cascades Nowak & Sigmund (1993). The mapping of trade-offs among efficiency, robustness, and exploitation is based on a wide spectrum of memory-1 strategies. In accordance with modern classifications of the strategy space Hilbe et al. (2015), this includes deterministic baselines (All-C/All-D, TFT, WSLS, Grim) at one end, and continuous families of stochastic policies (e.g., ZD) at the other.

Table 1: Strategies of $2 \times 2$ general-sum games

| Type | Representative | Memory | Key trait |
|---|---|---|---|
| Deterministic | All-C/All-D | 0 | Baseline references for extreme behaviors |
| Deterministic | TFT | 1 | Strong cooperation incentive, fragile to error cascades |
| Deterministic | WSLS | 1 | Self-corrects under noise |
| Deterministic | Grim | $\infty$ | Strong deterrence, extremely brittle to errors |
| Stochastic | Generous TFT | 1 | Forgiveness enables recovery of cooperation |
| Stochastic | ZD | 1 | Linear constraints on long-run payoffs (useful for exploitation analysis) |

## D EXTEND EXPERIMENTS

Here, we show regret and upper bound with 3 decomposed terms in winwin game (one of the $2 \times 2$ general-sum games).

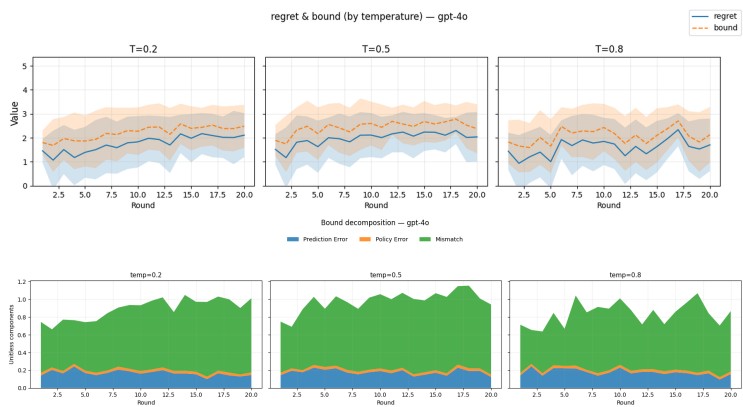

Figure 10: gpt4o's regret and upper bound in winwin game; opponent:random (episode:20, round:20)

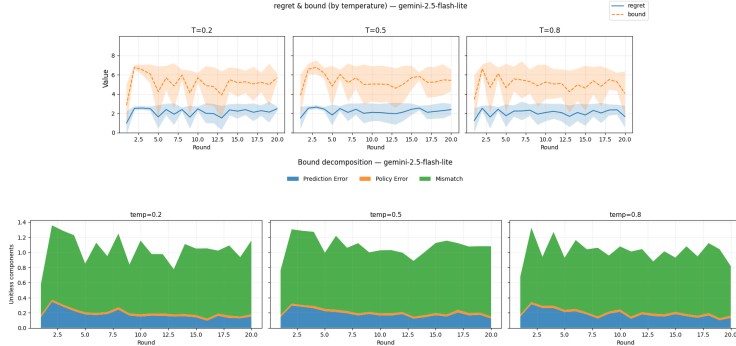

Figure 11: gemini2.5 flash lite's regret and upper bound in winwin game; opponent:random (episode:20, round:20)

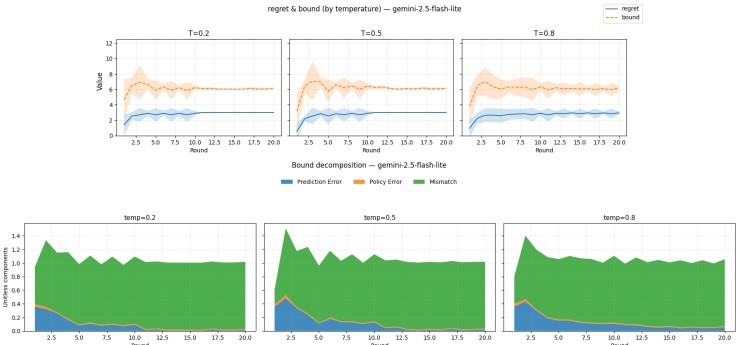

Figure 12: gemini2.5 flash lite's regret and upper bound in winwin game; opponent:tft (episode:20, round:20)

Here, we use GPT4o and gemini2.5 flash lite as LLMs (temperature:0.2, 0.5, 0.8). As shown in figure 12, prediction error decrease as the rounds go on. This tendency is also observed in prisoner's dilemma. In contrast, policy mismatch is dominant in winwin game. This means that models cannot select their action rationally. In fact, cooperation is the optimal strategy in winwin game, yet both models show the proportion choosing cooperation ranging from 15 to 30 %.

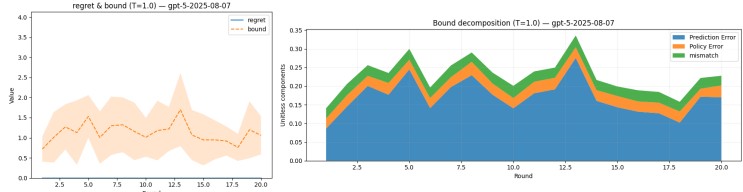

Figure 13: GPT5's regret and upper bound in winwin game; opponent:random (episode:10, round:20)

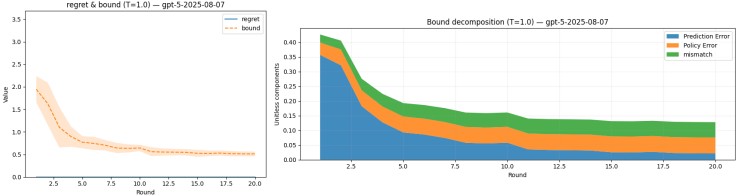

Figure 14: gpt5's regret and upper bound in winwin game; opponent:tft (episode:10, round:20)

when opponent strategy is TFT, prediction error decrease similarly. However, in GPT5 prediction error is dominant unlike gpt4o and gemini2.5 flash lite as shown in figure 13. In fact, GPT5 consistently chooses cooperative behavior 100%. This result reinforces that GPT5 makes rational choices.

## E    LLM USAGE

I used a LLM to refine English expressions.

