# OpenReview forum: "REAL-TIME RISK EVALUATION FOR LLM DECISION- MAKING VIA AN REGRET BOUND"
_ICLR.cc/2026/Conference — ICLR 2026 Conference Withdrawn Submission_

### Official Review · Reviewer_BKcF · 2025-10-15

**Soundness:** 1
**Presentation:** 1
**Contribution:** 1
**Rating:** 0
**Confidence:** 5

**Summary:**

This paper proposes a per-round regret decomposition for LLM-based agents into prediction error, policy error, and policy mismatch, claiming it provides a real-time “risk evaluation” for LLM decisions. The authors test it on simple 2×2 games (Prisoner’s Dilemma, Ultimatum, Win-Win) using GPT-based models.

**Strengths:**

The topic could become interesting if extended to non-toy settings, real online environments, or theoretically justified forms of dynamic regret analysis.

**Weaknesses:**

1. **Technical Depth**
   * The final result in Eq 6 is a direct restatement of standard Lipschitz and triangle inequalities from classical online learning.

2. **Conceptual and Mathematical Confusion**
   * The decomposition is just an additive partition of regret into L₁ distances; it does not represent a new bound or performance measure.
   * Regret is inherently non-decreasing over time, yet all figures show decreasing regret curves—this indicates either a misunderstanding or incorrect implementation. They define per-round regret, I am not sure this is well-defined and also plotted in the graph

3. **Not Clear Experiments**
   * Figures lack numerical scales, explanations, or statistical comparisons. The discussion simply narrates obvious behavioral differences (“GPT5 defects less”) without analysis.
   * The claimed “diagnostic value” of the decomposition is never demonstrated; the results provide no actionable insight or correlation with performance.
   * My main point is that the experiments use only static, toy 2×2 games with hard-coded payoff matrices and no adaptive or learning component. Hence, the results do not provide new insights into LLM performance or regret minimization. A constructive suggestion would be to test the proposed framework on more complex or dynamic environments where prediction and policy mismatch could vary meaningfully over time. Also please add the prompt that authors used. Some of the contents in Appendix should be moved to main paper.

4. **Poor Writing and Presentation Quality**
   * The paper has many grammatical errors and awkward phrasing (“an regret bound,” “LLMs tends to choose defect”).
   * Notation is inconsistent and undefined in several places. -- For example, name of the LLM is not exactly same as the official name and many inconsistency between figure and main paper (GPT4o, gpt-4o), inconsistent upper case and lower case.  No definition on Y, A, policy map, etc...
   * Figures are unlabeled, and captions are uninformative, and not easy to read.

**Questions:**

Written in Weaknesses

---

### Official Review · Reviewer_oco5 · 2025-10-21

**Soundness:** 3
**Presentation:** 2
**Contribution:** 1
**Rating:** 2
**Confidence:** 4

**Summary:**

This paper studies how to quantify per-round regrets for LLM decision making. The authors derive upper bounds for two-player LLM games. Then, they conduct experiments in a variety of games with multiple LLMs, and compare the derived upper bound and the real regret bound.

**Strengths:**

Quantifying LLM decision-making regret is an important topic. The setup and frequently used notation are explained clearly. Experiment results are provided to verify their theoretical founding.

**Weaknesses:**

The contribution of this paper is limited and I believe it does not match the bar of this conference. Although the authors spend much efforts in deriving the upper bounds for regret in the main text, the result is a direct corollary by applying basic triangular inequalities, and therefore, is trivial.

Besides, empirically, it seems the gap between derived upper bound and the real regret is very large (e.g., Fig 1 and 7), which suggests the derived upper bound is not an effective indicator. It is valuable to investigate tighter upper bounds to characterize the real-time risk.

**Questions:**

In experiments, given that the agents are LLM, it is not straightforward for me to see how the upper bound (Eq. 6) is computed. For example, how to compute $\hat{\mu}_t$ and $f(\hat{\mu}_t)$? I would suggest the authors to elaborate this in main text (or move the discussion from appendix to the main text)

---

### Official Review · Reviewer_YBPU · 2025-10-27

**Soundness:** 1
**Presentation:** 1
**Contribution:** 1
**Rating:** 2
**Confidence:** 3

**Summary:**

This paper aims to provide a decomposition for the per-round regret, which might offer a diagnostic tool for deploying LLM agents. Authors give upper bounds of each part of the decomposition.

**Strengths:**

Providing a practical diagnostic tool for deploying LLM agents is a timely topic.

**Weaknesses:**

- Authors hope to provide a diagnostic tool for analyzing the per-round regret. However, the final three terms of the decomposition depends on the opponent's policy, which cannot be known in many cases.

- This manuscript seems not to be ready for the submission since the presentation is highly uncelar, which makes readers hard to follow. Moreover, there are many typos across the whole manuscript. The writing of the abstract is distracting. It looks like authors hope to formally define many definitions in the abstract, which makes no sense to me. Even in this case, authors still do not define what is $\mathbb{Y},\mathbb{A}$ in the abstract.

- The presentation of the problem setup in Section 3 is chaotic. Authors do not justify the reference rule. Many typos appear in section 3, which makes readers hard to comprehend.

**Questions:**

I would suggest authors carefully revising the manuscript. For example, authors should present main message in the abstract rather than trying to formally write the problem setup there.

---

### Official Review · Reviewer_GEfJ · 2025-10-30

**Soundness:** 3
**Presentation:** 2
**Contribution:** 2
**Rating:** 4
**Confidence:** 3

**Summary:**

This paper introduces a novel framework for real-time risk certification for Large Language Model (LLM) agents operating in interactive environments, aiming to upper-bound the per-round regret. The core contribution is the derivation of this regret bound, which is decomposed into three specific terms: prediction error, policy error, and policy mismatch. This decomposition yields real-time diagnostics that crucially allow attribution of decision risk, revealing whether failures stem from flawed opponent modeling (belief calibration) or from suboptimal action selection given the agent's belief. Experiments conducted across $2 \times 2$ general-sum games (like the Prisoner's Dilemma) and Ultimatum games demonstrate that the dominant risk component shifts depending on the specific LLM, game, and opponent strategy, thus establishing per-round online certification as a practical diagnostic tool for safer and more effective LLM agent deployment.

**Strengths:**

* The paper introduces a novel per-round regret certificate to set an instantaneous upper bound on LLM decision risk in interactive environments.

* The framework adapts classical per-round bounds from online learning, ensuring a rigorous theoretical foundation that is fully computable in real time.

* Experiments validate the framework across key strategic settings, including $2 \times 2$ general-sum games (like Prisoner's Dilemma) and the Ultimatum Game, utilizing multiple state-of-the-art LLMs (GPT4o, GPT5, gemini2.5 flash lite).

**Weaknesses:**

* The experimental validation utilizes only a **fixed set of payoff parameters** for each specific game structure (e.g., Prisoner's Dilemma, Ultimatum Game). This critically limits the verification of the framework's robustness, as the key calculated components—specifically the regret bound's scaling factor ($L$) and the optimal action choice ($\pi^*_t$) for the Policy Error term—are fundamentally dependent on those specific payoff values.

* The framework critically relies on real-time access to the opponent's true action distribution ($\mu_t$) and payoffs ($Q_t$). While standard in theory, this assumption limits its practical utility in real-world black-box settings, as $E_{pred}$ (Prediction Error) and the optimal response ($\pi^*_t$) cannot be accurately computed without knowing $\mu_t$.

**Questions:**

* To validate the framework's robustness, are supplementary experiments necessary using diverse payoff matrices that systematically vary the domain-dependent scaling factor ($L$)?
* If real-world deployment prohibits accessing the opponent's true distribution ($\mu_t$), how does the framework maintain its diagnostic utility and theoretical guarantees under partial observability?

---

### Note · Authors · 2025-12-02

I have read and agree with the venue's withdrawal policy on behalf of myself and my co-authors.